# Validation of Generalized Anxiety Disorder 6 (GAD-6)—A Modified Structure of Screening for Anxiety in the Adolescent French Population

**DOI:** 10.3390/ijerph20085546

**Published:** 2023-04-17

**Authors:** Anja Todorović, Cédric Baumann, Myriam Blanchin, Stéphanie Bourion-Bédès

**Affiliations:** 1UR4360 APEMAC, School of Public Health, Faculty of Medicine, University of Lorraine, 54000 Nancy, France; 2Methodology, Data Management and Statistics Unit, University Hospital of Nancy, 54000 Nancy, France; 3SPHERE U1246, University of Nantes, University of Tours, INSERM, 44000 Nantes, France; 4Versailles Hospital, University Department of Child and Adolescent Psychiatry, 78157 Versailles-Le-Chesnay, France

**Keywords:** Generalized Anxiety Disorder, GAD-7, GAD-6, anxiety, adolescence, Differential Item Functioning, self-reported health status

## Abstract

Anxiety disorders remain underdiagnosed and undertreated, especially in child and adolescent populations. This study aimed to examine the construct validity of the Generalized Anxiety Disorder Scale 7 (GAD-7) in a sample of French adolescents by combining the Classical Test Theory (CTT) and the Item Response Theory (IRT) and to assess the invariance of items. A total of 284 adolescents enrolled in school in the Lorraine region were randomly selected to participate in a cross-sectional study. A psychometric evaluation was performed using a combination of CTT and IRT analyses. The study of psychometric properties of GAD-7 revealed poor adequation to the sample population, and engendered the deletion of one item (#7) and the merger of two response modalities (#2 and #3). These modifications generated the new GAD-6 scale, which had a good internal consistency reliability (Cronbach α = 0.85; PSI = 0.83), acceptable goodness-of-fit indices (χ2 = 28.89, df = 9, P = 0.001; RMSEA (90% CI) = 0.088 [0.054; 0.125]; SRMR = 0.063; CFI = 0.857), and an acceptable convergent validity (r = –0.62). Only one item (#5) had a consistent Differential Item Functioning (DIF) by gender. This study evaluated the structure of the GAD-7 scale, which was essentially intended at discriminating adolescent patients with high levels of anxiety, and adapted it to a population of adolescents from the general population. The GAD-6 scale presents better psychometric properties in this general population than the original GAD-7 version.

## 1. Introduction

Adolescence is a crucial time of development and growth, occurring between the ages of 10 and 19 years old, when youngsters learn how to manage their emotions and develop coping mechanisms [1]. Children and adolescents with anxiety tend to experience excessive worries about their performance at school or about their competence [2]. Furthermore, the presence of an anxiety disorder has been found to increase the risk of suicide attempts among adolescents [3,4]. The World Health Organization (WHO) estimates anxiety to be the sixth leading cause of illness in adolescents aged 10–14 years old, and the ninth in those aged 15–19 [1], while the Chartres Study estimates the prevalence of emotional disorders among French 8–11-year-olds to be 5.9% [5]. Despite their prevalence rates, anxiety disorders remain underdiagnosed and undertreated, especially in the child and adolescent populations [6].

In the context of the Coronavirus disease 2019 (COVID-19) pandemic and lockdown, children and adolescents have been shown to be highly vulnerable to emotional disturbances and adverse mental health effects [7]. In China, the prevalence of anxiety symptoms among high school students was estimated to be 37.4% during the COVID-19 outbreak [8]. The PIMS-CoV19 study (Feelings and Psychological Impact of the COVID-19 Epidemic Among Students in the Grand Est Area, France) revealed that 15.2% of university students experienced moderate anxiety symptoms, 9.8% experienced severe anxiety [9], and 22% experienced high levels of perceived stress [10]. 

According to the Fifth Edition of the Diagnostic and Statistical Manual of Mental Disorders (DSM-V), the diagnostic criteria for Generalized Anxiety Disorder (GAD) include excessive feelings of anxiety and worry, as well as the avoidance of activities with possible negative outcomes, occurring on more days than not for 3 months or longer, and often associated with symptoms such as restlessness and/or muscle tension [2,11]. 

In clinical research and primary care, emotional troubles such as Generalized Anxiety Disorder are typically screened for with self-administered questionnaires. The GAD-7 scale (Generalized Anxiety Disorder 7-item scale) [12] is increasingly used in screening, as well as in clinical research, and was proven to exhibit sound psychometric properties in the assessment of various groups of people including adults and adolescents, both in the general population and in patients with anxiety disorders [13,14,15,16,17]. However, the measurement properties have not been demonstrated in the French youth population. Recent studies have shown increasing interest in validating GAD-7 and other self-assessment questionnaires in adolescent and pediatric populations [13,15,18,19,20,21,22] in different languages, as concern grows for the mental well-being of youngsters and as early detection is considered an important factor in the successful treatment of mental disorders. Adolescents could potentially have a different understanding of certain items when compared to their adult counterparts, either due to a less developed vocabulary or due to a different generational perception of certain terms. This is why it is important to systematically test the psychometric properties in these populations, in order to dispose of a valid and robust screening tool. 

To study psychometric properties such as internal consistency, factor structure, and convergent validity, traditionally, the Classical Test Theory (CTT) method is used. It is based on the theory of a “true score”, which represents the latent trait intended to be measured, which in this case is anxiety [23]. More recently, the Item Response Theory (IRT) method has been used in numerous validation studies as a way to address measurement challenges in psychometric scales. The IRT method is model-based, with the goal of estimating parameters for each item in a scale, separating the person’s responses to the items from their underlying level of the latent trait that is being measured by the scale [24]. A limitation of the CTT method is that it does not consider the ability of the individual responding to the items, nor the difficulty of the items that make up the scale. On the contrary, individual’s ability and item difficulty are properties which are the basis of the IRT method. Therefore, in recent publications, the modern method based on Item Response Theory is used to complement the results of the classical method [25,26]. Another quality of measure of a questionnaire is the invariance of items, meaning that items should only be sensitive to the level of the latent trait, and not to other characteristics, such as age or gender. To detect potential measurement invariance of scale items, the most commonly used method in the literature is Differential Item Functioning (DIF). DIF represents statistical proof that an item tends to perform differently in a particular group of the sample when compared to the reference group [27]. In the context of IRT, DIF implies a difference in the item response function, connected to a third variable that often indicates group membership [27]. Ideal conditions imply that the pattern of responses should not vary across groups that have the same level of the latent trait, and the presence of a DIF indicates that the measurement invariance criterion has not been met [28]. 

Given the recent works on the GAD-7 among adolescents in different languages [13,15,18,19,20,21,22], the need for a brief and robust instrument to screen anxiety among French adolescents, as well as the importance of verifying the structure stability of GAD-7 during a particularly anxiety-inducing time, such as the COVID-19 pandemic, this study aimed to achieve the following:(a)Examine the construct validity of the French version of the GAD-7 in a sample of French adolescents (11–18 years old) by combining the Classical Test Theory (CTT) and the Item Response Theory (IRT).(b)Explore the invariance of items on the French version of the GAD-7 scale.

## 2. Methods

### 2.1. Study Design and Procedure

This is an ancillary study to the PIMS2-CoV19 study. PIMS2-CoV19 study (Feelings and Psychological Impact of the COVID-19 Epidemic Among Students in the Grand Est Area, France) is an observational and cross-sectional study that was conducted in France (Grand Est area) from 26 May to 6 July 2020, following the first French national lockdown, via an online survey that was accessible with an internet connection. The PIMS2-CoV19 study aimed to assess the effects of the COVID-19 pandemic on children and adolescents.

Schools were selected at a random from an exhaustive list of middle and high schools attached to the Nancy-Metz Academy (Lorraine region), including 408 middle and high school establishments and 184,987 students over the academic year of 2019/2020. The selection was performed using proportionate stratification for baseline ascertainment and recruitment. In selected establishments, the survey was diffused through institutional mailing lists, instructing children and parents to access a link containing instructions and information concerning their participation in the study. The survey included the following elements: an informed consent form, questions about demographics, living, and learning conditions during the lockdown, anxiety [12,16], and Health-related Quality of Life (HRQoL) [29] questionnaires. 

All data were obtained at the time of the online survey and their anonymous nature did not allow us to trace back personal data. The study was conducted in full compliance with national regulations and the principles of the Declaration of Helsinki. The study protocol was approved by the National Commission on Informatics and Liberty (Comité National Informatique et Liberté, registration 2220408).

### 2.2. Participants

All adolescents enrolled in grades 6 through 12 in a school in the Nancy-Metz Academy (Lorraine region, France) were eligible to participate in the study. The sample included 284 consenting adolescents (11–18 years old) from 230 distinct households. As presented in Table 1, more than half of the adolescents were female (55.3%). The average age was 14.5 years (SD = 3.1) and 60.2% of pupils were enrolled in a middle school. Of the 39.8% of the sample that was being educated in a high school, 72.4% were enrolled in a general high school. Only 8.1% of participants were only children.

### 2.3. Measures

#### 2.3.1. Sociodemographic Characteristics and Living Conditions

Other data collected included adolescents’ age, sex, education level (middle school/high school), and living and learning conditions during lockdown, as well as data concerning the presence of people infected by SARS-CoV-2 in their immediate environment.

#### 2.3.2. Anxiety

Anxiety was assessed by the Generalized Anxiety Disorder-7 items scale (GAD-7), which was developed based on DSM-IV diagnostic criteria [30]. Participants were asked how often they were bothered by anxiety symptoms during the last two weeks [12]. Each item is scored on a 4-point Likert scale: “not at all”, “several days”, “more than half the days”, and “nearly every day”, scored as 0, 1, 2, and 3, respectively [12]. Thus, GAD-7 scores range from 0 to 21, with scores of ≥5, ≥10, and ≥15 representing mild, moderate, and severe anxiety symptom levels, respectively [12]. The French version of the GAD-7 questionnaire was developed through a forward–backward translation by two independent native French speakers and two independent native English speakers and was only validated in an adult French population [16]. The French version of the GAD-7 yields a valid and reliable clinical assessment of anxiety among the adult population [16]. 

**Table 1 ijerph-20-05546-t001:** Sociodemographic characteristics of the adolescent sample population during lockdown.

	N = 284 N (%)/Mean (SD)
Age	14.5 (3.1)
Gender	
Boy	127 (44.7)
Girl	157 (55.3)
Grade	
6th	43 (15.1)
7th	43 (15.1)
8th	37 (13.0)
9th	48 (16.9)
10th	42 (14.8)
11th	40 (14.1)
12th	31 (10.9)
Education level	
Middle school	171 (60.2)
High school	113 (39.8)
High school type	
General high school	89 (72.4)
Technological high school	17 (13.8)
Vocational high school	17 (13.8)
Number of siblings	
Zero	23 (8.1)
One or more	261 (91.9)

Abbreviation: SD (standard deviation).

#### 2.3.3. Self-Perceived Health Status

The self-perceived health status was assessed using the KIDSCREEN-27 questionnaire [29]. It is a quality of life questionnaire derived using psychometric methods from the longer KIDSCREEN-52 questionnaire [31], composed of 27 items, divided into 5 dimensions evaluating the following: physical well-being (5 items), psychological well-being (7 items), parent relations and autonomy (7 items), social support and peers (4 items), and school environment (4 items) [29]. The responses are presented in the form of a 5-point Likert scale, assessing the frequency of certain feelings as “never” (1), “seldom” (2), “quite often” (3), “very often” (4), and “always” (5), or the intensity as “not at all” (1), “slightly” (2), “moderately” (3), “very” (4), and “extremely” (5), during the past week [29]. Scores are coded from 1 to 5, negatively formulated items are recoded, and higher summed scores indicate better HRQoL. Rasch scores are computed for each dimension and transformed into values with a mean (SD) of 50 (10). Normative values are available for adolescents from 11 European countries [32]. Answering the KIDSCREEN-27 questionnaire requires 10–15 min.

### 2.4. Statistical Analyses

#### 2.4.1. Descriptive Analyses

Data are represented as numbers and percentages for categorical variables and as the means ± standard deviations (SDs) for continuous variables. To compare variables between groups, Fisher’s exact test was used for qualitative variables, and Student’s *t*-test was used for quantitative variables. The results were considered significant when the *p*-value was < 0.05. Furthermore, we examined the presence of floor and ceiling effects in GAD-7 and KIDSCREEN-27 overall scores.

#### 2.4.2. Primary Analyses

A combination of Classical Test Theory (CTT) and Item Response Theory (IRT) was applied to examine the psychometric properties of the GAD-7 scale. The statistical analyses were performed using SAS 9.4 software (SAS Inst., Cary, NC, USA) and the ROSALI module in Stata [33].

The structure of the GAD-7 questionnaire was studied using a confirmatory factor analysis (CFA), which was performed using covariance-based structural equation modelling with maximum likelihood estimation. CFA was performed to test the adequacy to the predefined unidimensional factor structure model. The adequacy was evaluated from different indices, including the root mean square error of approximation (RMSEA), standardized root mean residual (SRMR), and comparative fit index (CFI). The models were judged as good if RMSEA < 0.08, SRMR ≤ 0.07, and CFI > 0.9 [34,35]. The internal consistency reliability was evaluated by Cronbach’s alpha coefficient (α), which was considered good from 0.8 to 0.89, and excellent if >0.9 [36].

Convergent validity is evaluated by searching for correlations between two instruments measuring the same concept [37]. The correlations between GAD-7 and the psychological well-being dimension of KIDSCREEN-27 were assessed using Pearson’s correlation coefficient (r). These two assessment tools measure opposite concepts (i.e., anxiety and psychological well-being); therefore, a negative correlation would be expected in order to establish convergent validity. Convergent validity is generally considered adequate if correlation value is >0.50 [38], and inversely if it is negative: <−0.50.

The Partial Credit Model (PCM) is the IRT model and Rasch family model that was used to confirm the unidimensionality of the scale. The Person Separation Index (PSI) was calculated as an indicator of internal consistency reliability. PSI was considered as acceptable if >0.8, and individual item fit residual statistics were acceptable when the value ranged from −2.5 to 2.5 [39].

#### 2.4.3. Secondary Analyses

The DIF analysis was performed partly using the ROSALI algorithm [33,40] based on Partial Credit Models. To detect differences in item difficulty parameters, a PCM estimating different item difficulty parameters for all items between the two groups is compared to a PCM assuming no DIF, i.e., equal item difficulty parameters between groups using a likelihood ratio test. If the test is not significant, no DIF is assumed. Otherwise, items where DIF is suspected are identified one-by-one in an iterative step by relaxing equality constraints of item difficulty parameters item-by-item. The final PCM model estimates the size of group effect, adjusted for DIF if appropriate, and the type and size of DIF.

## 3. Results

### 3.1. Sociodemographic, Living, and Learning Characteristics

The analysis included 284 consenting adolescents (11–18 years old) from 230 distinct households. As presented in Table 2, more than half of the adolescents were female (55.3%). The average age was 14.5 years (SD = 3.1) and 60.2% of pupils were enrolled in a middle school. Of the 39.8% of the sample that was being educated in a high school, 72.4% were enrolled in a general high school. During the lockdown, 65.2% of the sample lived with both parents, and 59.2% reported living in a rural environment. While 83.1% declared a house as their accommodation type, 82.0% had access to a garden or land for exclusive use. Of the sample, 21.6% reported never going outside during this period. The average time spent on schoolwork was 2 hours per day (SD = 0.9). Only 3.6% had one or more confirmed cases of COVID-19 within their accommodation, while 28.5% had at least one confirmed case within their entourage.

### 3.2. Anxiety and Self-Perceived Health

The adolescents’ mean GAD-7 scores per item are shown in Table 3. Girls scored significantly higher than boys on the mean overall GAD-7 score (*p* = 0.0004), as well as on item scores for items one (*p* = 0.0078), two (*p* = 0.0002), three (*p* = 0.0001), four (*p* = 0.0038), and seven (*p* = 0.0438). No statistically significant difference was detected between middle schoolers and high schoolers.

The average overall score is presented in Table 4, with the mean score being 4.2 (SD = 4.4). A floor effect was observed (20.4%). 

As shown in Table 5, 63.7% of adolescents had normal anxiety levels, 23.9% had mild anxiety, 8.5% had moderate anxiety, and 3.9% had severe anxiety levels.

The mean self-reported quality of life scores per dimension are shown in Table 6. The mean score for the psychological well-being dimension was 47.2 (SD = 10.1). The lowest mean score was observed for the peers and social support dimension at 36.9% (SD = 13.1), and the highest was for the autonomy and parents dimension at 48.6 (SD = 12.2). No floor or ceiling effect was detected.

### 3.3. Dimensionality and Other Psychometric Properties of the GAD-7

When testing the unidimensionality of the GAD-7 scale, a confirmatory factor analysis showed high factor loadings for items #1 (1.00), #2 (0.88), #3 (1.03), #4 (0.87), and #6 (0.81); however, factor loadings were low for items #5 (0.58) and #7 (0.62). The goodness-of-fit indices were lower than acceptable, with χ2 = 34.64, df = 14, *p* = 0.002, and RMSEA (90% CI) = 0.072 [0.042; 0.103], SRMR = 0.111, and CFI = 0.739. The internal consistency was good, with Cronbach’s alpha at α = 0.87. A negative correlation between the GAD-7 score and KIDSCREEN-27 psychological well-being dimension was found, with Pearson’s correlation coefficient estimated at r = −0.61 (*p* < 0.0001), confirming an acceptable convergent validity. 

The internal consistency reliability was reassessed with Person Separation Index, which was acceptable at PSI = 0.83. The individual item fit residual statistics for items #5 and #7 were less good, with standardized infit statistics at 3.948 and 2.659, respectively. 

As shown in Figure 1a, no significant overlap was observed between the information curve and the curve representing the density of the latent trait. Furthermore, the figure indicates overlaps between thresholds of response modes, specifically for modes #2 (“more than half the days”) and #3 (“nearly every day”). In addition, the positions of response thresholds for item #7 (“Feeling afraid as if something awful might happen”) indicate an overall increased difficulty for this item within this population, as well as a small discrimination power, as shown in Figure 1b.

### 3.4. Dimensionality and Other Psychometric Properties of the GAD-6 Version

To create the new GAD-6 scale, two changes to the initial GAD-7 scale have been made. Firstly, item #7 was deleted, and secondly, response modes 2 (“more than half the days”) and 3 (“nearly every day”) have been merged into one response mode 2 (“nearly every day”).

When testing the unidimensionality of the new GAD-6 scale, a confirmatory factor analysis showed high factor loadings for all items (0.83–0.92), except item #5 (0.65). The goodness-of-fit indices were better with this structure, with χ2 = 28.89, df = 9, *p* = 0.001, and RMSEA (90% CI) = 0.088 [0.054; 0.125], SRMR = 0.063, and CFI = 0.857. The internal consistency was good, with Cronbach’s alpha at α = 0.85. A negative correlation between the GAD-7 score and KIDSCREEN-27 psychological well-being dimension was found, with Pearson’s correlation coefficient estimated at r = −0.62 (*p* < 0.0001), showing an acceptable convergent validity. Furthermore, a residual covariance between items #2 and #3 has been observed (χ2 = 14.72, df = 8, and RMSEA = 0.054, SRMR = 0.037, and CFI = 0.952).

The internal consistency reliability of the new GAD-6 scale was reassessed using the Person Separation Index, which was acceptable at 0.83. The individual item fit residual statistics were within allowed limits (±2.5), except for item #5 (“being so restless that it’s hard to sit still”), which had its standardized infit statistics at 4.242. 

As shown in Figure 2, significant overlap has been observed between the information curve and the curve representing the density of the latent trait when testing the new GAD-6 scale. Furthermore, the problem of the response threshold overlap has been resolved with this new structure.

**Figure 2 ijerph-20-05546-f002:**
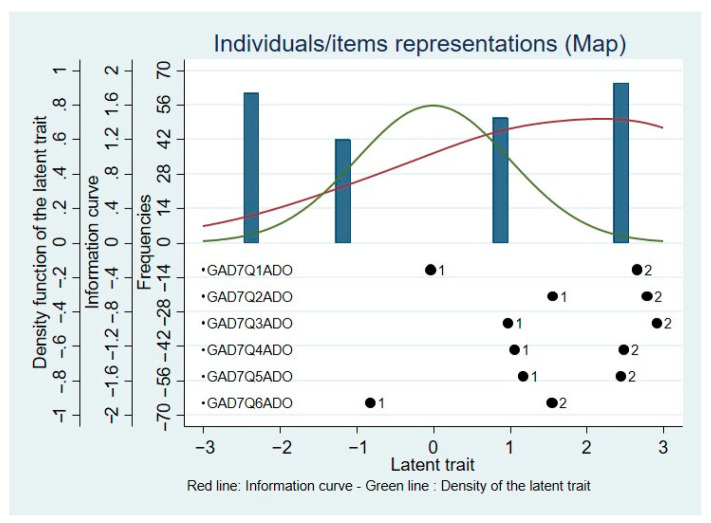
Map of individuals/items representations: latent trait (anxiety) in function of density of the latent trait, information curve, and frequencies, with indicated thresholds of response modes for each item (GAD-6).

### 3.5. Differential Item Functioning of the GAD-6 Items

No DIF was detected for any of the items when tested for differences between middle schoolers and high schoolers. 

When tested for differences between boys and girls, a group effect was revealed when processing Step B, estimated at 0.79 (SE = 0.26, *p* = 0.0023). The likelihood ratio test was significant at χ2 = 28.48 (df = 11, *p* = 0.0027), which confirmed the presence of a DIF. 

No DIF was detected for items #1, #2, #3, #4, and #6 when tested for differences between boys and girls. However, for item #5 (“being so restless that it’s hard to sit still”), the gender group showed statistically significant (χ2 = 14.012, df = 2, *p* = 0.0009) uniform DIF, with its test of uniform difference χ2 = 0.31 (df = 1, *p* = 0.58) not showing significant results. Results showed higher item difficulty estimations for girls (δj_1 = 2.15 (SE = 0.28); δj_2 = 3.39 (SE = 0.35)) than for boys (δj_1 = 1.16 (SE = 0.28); δj_2 = 2.40 (SE = 0.35)), indicating that at the same level of anxiety, boys were more likely to report “being so restless that it is hard to sit still” than girls. 

Once the group effect was adjusted for the DIF, the coefficient of the group effect was estimated at 0.96 (SE = 0.27; χ2 = 12.82; *p* < 0.001).

## 4. Discussion

This study aimed to examine the construct validity of the French version of the GAD-7 in a sample of French adolescents (11–18 years old) by combining the Classical Test Theory (CTT) and the Item Response Theory (IRT). Furthermore, this study also aimed to explore the invariance of items on the French version of the GAD-7 scale. 

The results showed that this sample population was generally not experiencing severe symptoms of anxiety, with 63.7% reporting “normal levels” of GAD. Moreover, a floor effect of 20.4% was observed for the overall GAD-7 score in this sample, indicating that the French version of the GAD-7 does not have great discriminating properties in the lower overall scores. Mossman et al. [13] found that the GAD-7 scores may be used to assess anxiety symptoms and to differentiate between mild and moderate GAD in adolescents presenting with GAD, and Löwe et al. [17] found good validity for screening in the general adult population. Considering that this study was conducted within a general population of adolescents presenting with normal-to-mild symptoms of GAD, the floor effect was the first indicator that the French version of the GAD-7 may not be well adapted to screening for GAD within this particular population. Furthermore, the cut-off values for differentiating between levels of anxiety need to be reexamined in this population (Figure 1), as response thresholds might not be transposable from the adolescent GAD patient population to the general adolescent population.

The results of the CFA showed good internal consistency reliability for GAD-7 (α = 0.87); however, they were slightly lower in comparison with previous results [12,15,16,17]. On the contrary, the goodness-of-fit indices were very low (RMSEA (90% CI) = 0.072 [0.042; 0.103], SRMR = 0.111, and CFI = 0.739), indicating that the proposed model does not fit the data adequately. Additionally, two major issues raised concerns over the structure of the French version of GAD-7 in this population: (a)An increased item difficulty was observed for item #7 (“Feeling afraid as if something awful might happen”) in this population-based sample (Figure 1a), with the standardized infit value being higher than recommended for this item at 2.659. Furthermore, as shown in Figure 1b, item #7 demonstrated a small discrimination power, which was possibly due to a decrease in its cultural relevance and ability to assess anxiety. It is probable that this item is less difficult and has a stronger discriminatory power in study samples with higher anxiety levels; however, it was clearly not a good fit for our dataset, due to overall low levels of anxiety.(b)Significant overlap was observed between response thresholds for response modes #2 (“more than half the days”) and #3 (“nearly every day”), as shown in Figure 1a. This is suspected to be caused by a semantic similarity between these two terms in French, as well as a possible difficulty among adolescents to clearly distinguish between these two denominations. A study in Ghana and Côte d’Ivoire revealed a lack of additional information provided by response mode #2 (more than half the days), indicating a finding similar to ours [41].

Following these results, certain modifications to the initial GAD-7 scale were made. Firstly, due to its increased difficulty and small discrimination power in this population-based sample, item #7 was removed. Secondly, due to their significant overlap and semantic similarity in the French version of the scale, the response modalities #2 and #3 were merged into one–#2 (“nearly every day”). The newly obtained scale was, consequently, named GAD-6. It conserved items #1, #2, #3, #4, #5, and #6 of the original GAD-7 scale, and was scored on a 3-point Likert scale: “not at all”, “several days”, and “nearly every day”, scored as 0, 1, and 2, respectively. Thus, GAD-6 scores range from 0 to 12. The cut-off values for determining anxiety levels with this new scoring would need to be determined with further investigation. It is important to note that these new thresholds are valid for use in the French adolescent general population only and should not be transposed into populations of primary care patients, as further validation would be necessary for such use.

When testing the structure of the newly created GAD-6 scale, several improvements were observed. GAD-6 had better psychometric properties when compared to the GAD-7: good internal consistency reliability (Cronbach α = 0.85; PSI = 0.83), acceptable goodness-of-fit indices (χ2 = 28.89, df = 9, *p* = 0.001; RMSEA (90% CI) = 0.088 [0.054; 0.125]; SRMR = 0.063; CFI = 0.857), and acceptable convergent validity (r = −0.62). The decision to remove item #7 resulted in adequate reliability indices and helped improve the instrument’s structure. Furthermore, the decision to merge the response modalities #2 and #3 resulted in an improved threshold structure, as shown in Figure 2. These findings showed that the new GAD-6 structure was better adapted to the sample population than the original GAD-7.

Moreover, the new GAD-6 structure revealed a significant residual covariance between items #2 (“Not being able to stop or control worrying”) and #3 (“Worrying too much about different things”). While both of these items address “worrying”, they evaluate different aspects of it, specifically the “control” and the “intensity” aspects. Therefore, the authors agreed that it was clinically justified to keep these two items because of their relevance in assessing anxiety.

Item #5 (“being so restless that it’s hard to sit still”) initially showed bad standardized infit statistics in both the GAD-7 and GAD-6 scales. Whereas there were strong arguments for the removal of item #7, on the contrary, item #5 had a level of difficulty adapted to this sample. Moreover, despite its elevated infit statistics, item #5 was judged to be clinically relevant by authors, and a choice was made to conserve it within the new structure. Furthermore, the elevated infit statistics of item #5 could potentially be explained by the presence of a significant uniform DIF on this item for the gender group. Evidently, a higher item difficulty made girls less likely to report “being so restless that it is hard to sit still” compared to boys. There may be a cultural and societal reason for this difference. Indeed, it is common in our society to consider boys as being more turbulent and restless than girls. These comments, heard at a very young age by children, could affect the perception that boys have of their unruly behavior and thus respond to this item with less difficulty than girls. To account for this DIF in the scoring of the GAD-6 scale, a modeling DIF approach should be used, such as the IRT purification method for item calibration and scoring in the presence of DIF [42], or the ROSALI module [33]. This type of approach would allow item #5, which is clinically relevant and important, to remain in the new GAD-6 scale structure while considering the presence of DIF for this item.

Some limitations of this study remain. Firstly, the survey was conducted online, as face-to-face interviews were not possible due to pandemic-related measures. This may have resulted in a respondent bias. Secondly, data were collected only from the Grand Est area, and a larger sample size coming from different French areas might be needed to generalize the results. Furthermore, no data were collected regarding racial or ethnic origin in this sample. In France [43] and in Europe in general [44], race or ethnic origin is considered sensitive data, and the provision of solid justification is necessary in order to be granted permission to collect it. According to the GDPR [45], it is considered that this type of data should not be collected unless there is an absolute necessity for it. In this study, the racial factor did not appear to be sufficiently important in our opinion to collect it and justify its collection. Moreover, although race is a factor influencing the level of stress and anxiety, no study has mentioned to our knowledge that this factor can impact the perception of the items or the scale structure. Additionally, after a review of the international literature aiming to assess the structural validity of GAD-7 in different populations, an absence of the racial factor in data collection is noticeable [13,15,18,19,20,21,22], affirming the argument that it would not have an impact on the perception of items or on the structure of the scale. However, this study demonstrates relevant information that needs to be carefully taken into consideration in future screening studies of the general population of French adolescents.

Concerning future analyses, precautions need to be taken to choose the appropriate scale structure and response thresholds, depending on the sample population. Lastly, changes in the structure of GAD-7 resulting in an alternative GAD-6 structure inevitably result in a lack of comparability between scores obtained by the two auto-questionnaires, at least until adequate cut-off values for anxiety levels are determined for the GAD-6 scale.

## 5. Conclusions

In conclusion, GAD-6 has overall better psychometric properties than GAD-7 within this population: it has good internal consistency reliability, with acceptable goodness-of-fit indices, and acceptable convergent validity. All items of the GAD-6 are clinically and culturally relevant and exhibit levels of difficulty adequate for this population. The response modalities have also been adapted to the general population of French adolescents.

## Figures and Tables

**Figure 1 ijerph-20-05546-f001:**
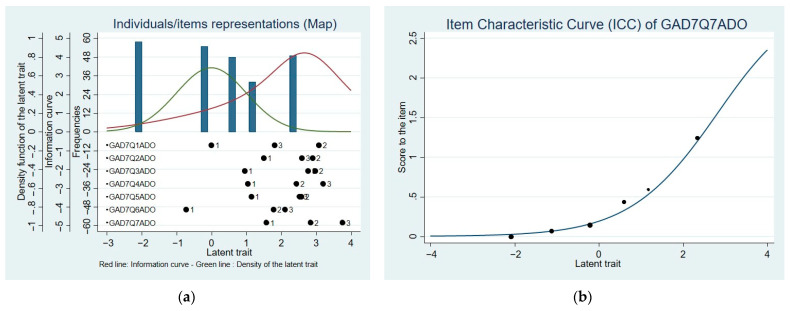
(**a**) Map of individuals/items representations: latent trait (anxiety) in function of density of the latent trait, information curve, and frequencies, with indicated thresholds of response modes for each item (GAD-7). (**b**) Figure 2 Item Characteristic Curve (ICC) of item #7 of GAD-7.

**Table 2 ijerph-20-05546-t002:** Sociodemographic, living, and learning characteristics of the adolescent sample population during lockdown (N = 284).

	N = 284 N (%)/Mean (SD)
Number of children or adolescents living in the family home	
One	46 (16.4)
Two	153 (54.4)
Three	62 (22.1)
Four	13 (4.6)
Five	7 (2.5)
One or more people in the home suffered from COVID-19	
Yes, confirmed case(s) requiring hospitalization	3 (1.1)
Yes, confirmed case(s) not requiring hospitalization	7 (2.5)
Yes, suspected case(s)	31 (10.9)
No	243 (85.6)
One or more people in the entourage suffered from COVID-19	
Yes, confirmed case(s) requiring hospitalization	21 (7.4)
Yes, confirmed case(s) not requiring hospitalization	60 (21.1)
Yes, suspected case(s)	29 (10.2)
No	174 (61.3)
Number of hours spent on schoolwork per day	2.0 (0.9)
Family structure	
Both parents	184 (65.2)
Other	98 (34.8)
Living environment	
Urban/semi-urban environment	115 (40.8)
Rural environment	167 (59.2)
Housing	
Apartment	48 (16.9)
House	236 (83.1)
Access to an exterior space	
For exclusive use	261 (91.9)
For collective use	8 (2.8)
No access to the outside	15 (5.3)
Exit outside the home	
Several times a day, almost every day	27 (9.6)
Once a day, almost every day	57 (20.2)
Several times a week, but not every day	45 (16.0)
Approximately once a week	35 (12.4)
Less than once a week	57 (20.2)
Never	61 (21.6)

Abbreviation: SD (standard deviation).

**Table 3 ijerph-20-05546-t003:** GAD-7 item scores (N = 284).

	Total N (%)	Boys N (%)	Girls N (%)	*p*	Middle School N (%)	High School N (%)	*p*
1. Feeling nervous, anxious, or on edge				0.0078			0.3251
Not at all	135 (47.5)	69 (54.3)	66 (42.0)		86 (50.3)	49 (43.4)	
Several days	111 (39.1)	49 (38.6)	62 (39.5)		67 (39.2)	44 (38.9)	
More than half the days	16 (5.6)	6 (4.7)	10 (6.4)		7 (4.1)	9 (8.0)	
Nearly every day	22 (7.7)	3 (2.4)	19 (12.1)		11 (6.4)	11 (9.7)	
2. Not being able to stop or control worrying				0.0002			0.3032
Not at all	197 (69.4)	104 (81.9)	93 (59.2)		125 (73.1)	72 (63.7)	
Several days	59 (20.8)	14 (11.0)	45 (28.7)		33 (19.3)	26 (23.0)	
More than half the days	15 (5.3)	6 (4.7)	9 (5.7)		7 (4.1)	8 (7.1)	
Nearly every day	13 (4.6)	3 (2.4)	10 (6.4)		6 (3.5)	7 (6.2)	
3. Worrying too much about different things				0.0001			0.0903
Not at all	177 (62.3)	93 (73.2)	84 (53.5)		115 (67.3)	62 (54.9)	
Several days	78 (27.5)	30 (23.6)	48 (30.6)		43 (25.1)	35 (31.0)	
More than half the days	17 (6.0)	4 (3.1)	13 (8.3)		9 (5.3)	8 (7.1)	
Nearly every day	12 (4.2)	0 (0.0)	12 (7.6)		4 (2.3)	8 (7.1)	
4. Trouble relaxing				0.0038			0.6811
Not at all	177 (62.3)	88 (69.3)	89 (56.7)		110 (64.3)	67 (59.3)	
Several days	70 (24.6)	27 (21.3)	43 (27.4)		41 (24.0)	29 (25.7)	
More than half the days	26 (9.2)	12 (9.4)	14 (8.9)		15 (8.8)	11 (9.7)	
Nearly every day	11 (3.9)	0 (0.0)	11 (7.0)		5 (2.9)	6 (5.3)	
5. Being so restless that it is hard to sit still				0.9543			0.5394
Not at all	182 (64.1)	83 (65.4)	99 (63.1)		106 (62.0)	76 (67.3)	
Several days	65 (22.9)	28 (22.0)	37 (23.6)		39 (22.8)	26 (23.0)	
More than half the days	20 (7.0)	8 (6.3)	12 (7.6)		15 (8.8)	5 (4.4)	
Nearly every day	17 (6.0)	8 (6.3)	9 (5.7)		11 (6.4)	6 (5.3)	
6. Becoming easily annoyed or irritable				0.0791			0.7566
Not at all	99 (34.9)	53 (41.7)	46 (29.3)		60 (35.1)	39 (34.5)	
Several days	113 (39.8)	49 (38.6)	64 (40.8)		69 (40.4)	44 (38.9)	
More than half the days	41 (14.4)	16 (12.6)	25 (15.9)		26 (15.2)	15 (13.3)	
Nearly every day	31 (10.9)	9 (7.1)	22 (14.0)		16 (9.4)	15 (13.3)	
7. Feeling afraid as if something awful might happen				0.0438			0.9522
Not at all	201 (70.8)	98 (77.2)	103 (65.6)		122 (71.3)	79 (69.9)	
Several days	59 (20.8)	23 (18.1)	36 (22.9)		35 (20.5)	24 (21.2)	
More than half the days	18 (6.3)	6 (4.7)	12 (7.6)		11 (6.4)	7 (6.2)	
Nearly every day	6 (2.1)	0 (0.0)	6 (3.8)		3 (1.8)	3 (2.7)	

**Table 4 ijerph-20-05546-t004:** Overall GAD-7 scores (N = 284).

	N	Mean	SD	Min	Q1	Median	Q3	Max	Skewness	Kurtosis	Floor Effect (%)	Ceiling Effect (%)
Overall GAD-7 score	284	4.2	4.4	0.0	1.0	3.0	6.0	19.0	1.4	1.6	20.4	-

Abbreviation: SD (standard deviation).

**Table 5 ijerph-20-05546-t005:** Levels of Generalized Anxiety Disorder in the sample population (N = 284).

Level of GAD	N (%)
Normal	181 (63.7)
Mild	68 (23.9)
Moderate	24 (8.5)
Severe	11 (3.9)

**Table 6 ijerph-20-05546-t006:** Overall KIDSCREEN27 scores per dimension (N = 284).

	N	Mean	SD	Min	Q1	Median	Q3	Max	Skewness	Kurtosis	Floor Effect (%)	Ceiling Effect (%)
Physical well-being	284	43.6	10.2	12.1	36.6	42.5	49.6	73.2	0.4	0.8	-	-
Psychological well-being	284	47.2	10.1	4.5	41.8	46.5	53.1	73.5	0.1	1.9	-	-
Autonomy and parents	284	48.6	12.2	1.7	40.6	46.5	55.8	74.4	0.3	0.7	-	-
Peers and social support	284	39.6	13.1	11.2	31.6	39.9	46.9	66.3	-0.1	-0.1	-	-
School environment	284	47.0	10.2	16.3	40.7	45.4	51.1	71.0	0.4	0.4	-	-

Abbreviation: SD (standard deviation).

## Data Availability

The data collected and analyzed during the current study are available from the corresponding author upon request.

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
