# Peer review of "Validation of Generalized Anxiety Disorder 6 (GAD-6)—A Modified Structure of Screening for Anxiety in the Adolescent French Population"

_ijerph, 2023, doi:10.3390/ijerph20085546_

Round 1

Reviewer 1 Report

The authors examined the validity of the Generalized Anxiety Disorder Scale among French adolescents. Overall, the manuscript is well written and my suggestions are as follows:

1) Introduction: (a) Please elaborate more on why this study is needed. (b) “the measurement properties have not been demonstrated in the French youth population.“ why is it important? Any evidence showing differences in validity between adolescents and adults as well as across cultures???

2) Methods: 2.4.2: Kindly specify the CFA estimation method.

3) Results: 3.4: Please justify both empirically and theoretically on the two major changes made to the scale.

4) Discussion: Any previous literature (especially among adolescents) that made changes to the GAD-7? Is this study in line/contradict with their findings??? Provide possible explanations for the inconsistency (if any).

5) Instead of relying on objective statistical results to decide on the removal/maintaining the items, this study also depends on the authors’ judgment. It would be great to explain this in detail (e.g., how may authors involved in making the decision, their expertise, justifications given).

Reviewer 2 Report

The topic researched in this article is of great interest. Unfortunately the data is lacking essential information, leading the results of this paper totally unreliable.

As it is well known, RACE plays a major role in feelings of depression, anxiety and stress. How do the authors account here for the large immigrant population who reside in France and have a nationality/racial based reason for anxiety? https://www.ined.fr/en/everything_about_population/demographic-facts-sheets/faq/how-many-immigrants-france/

In addition, what is the racial distribution of the adolescents in the Nancy area? Many people residing in Nancy are of North African or Middle_Eastern descent and therefore their approach to truthfully answering a survey may be very different due to their great distrust in the system. Many parents are educating their children to be distrusting of the system at large. Have interviewers been trained to address this aspect? Did the researchers send out interviewers belonging to the same ethnical/racial group to help reduce this type of a bias?

Since race has not been taken into account in this study, the results are not at all reliable and therefore the validity of this research is highly questionable.

Round 2

Reviewer 2 Report

This article needs to include a clear study limitation statement/paragraph in the limitation section. This statement should be reflecting on the lack of racial information and its potential to bias the results and, like the authors stated in their response to the reviewer, should also include the following sentences "In France [1] and in Europe in general [2], race or ethnic origin is considered as sensitive data, and it is necessary to provide solid justification in order to be granted permission to collect it. According to the GDPR, it is considered that this type of data should not be collected unless there is an absolute necessity for it.".

Author Response

We thank the reviewer for their time and effort in reviewing our manuscript.
